MicroCT illuminates the unique morphology of Shiinoidae (Copepoda: Cyclopoida), an unusual group of fish parasites

http://orcid.org/0000-0002-1769-8631 Bernot James P. 1 2 james.bernot@uconn.edu
http://orcid.org/0000-0001-8170-7734 Boxshall Geoffrey A. 3
http://orcid.org/0000-0001-5696-4333 Goetz Freya E. 2
http://orcid.org/0000-0003-4883-0022 Phillips Anna J. 2
1 Department of Ecology and Evolutionary Biology, University of Connecticut , Storrs, CT , United States
2 Department of Invertebrate Zoology, Smithsonian National Museum of Natural History , Washington, DC , United States
3 Department of Life Sciences, Natural History Museum , London , United Kingdom
Avenant-Oldewage Annemariè
Electronic publication date: 2024 Mar 4
Publication date: 2024
Volume: 12
Electronic Location ID: e16966
Received 2023 Nov 8; Accepted 2024 Jan 26
Copyright: © 2024 Bernot et al.
Copyright year: 2024
Copyright holder: Bernot et al.
License: This is an open access article distributed under the terms of the Creative Commons Attribution License, which permits unrestricted use, distribution, reproduction and adaptation in any medium and for any purpose provided that it is properly attributed. For attribution, the original author(s), title, publication source (PeerJ) and either DOI or URL of the article must be cited.
License URL: https://creativecommons.org/licenses/by/4.0/

Keywords: Copepod, Parasite, Functional morphology, MicroCT, Cyclopoida, Rostrum, Attachment, Fish, Scombridae, Scomberomorus

Funding: National Museum of Natural History, Smithsonian Institution NSF Postdoctoral Research Fellowships in Biology Program 2010898 Logistical support and equipment access was supported through National Museum of Natural History, Smithsonian Institution. This material is based in part upon work supported by the NSF Postdoctoral Research Fellowships in Biology Program under Grant No. 2010898 to James P. Bernot. The funders had no role in study design, data collection and analysis, decision to publish, or preparation of the manuscript.

==============================
The copepod family Shiinoidae Cressey, 1975 currently comprises nine species of teleost parasites with unusual morphology and a unique attachment mechanism. Female shiinoids possess greatly enlarged antennae that oppose a rostrum, an elongate outgrowth of cuticle that originates between the antennules. The antennae form a moveable clasp against the rostrum which they use to attach to their host. In this study, we use micro-computed tomography (microCT) to examine specimens of Shiinoa inauris Cressey, 1975 in situ attached to host tissue in order to characterize the functional morphology and specific muscles involved in this novel mode of attachment and to resolve uncertainty regarding the segmental composition of the regions of the body. We review the host and locality data for all reports of shiinoids, revise the generic diagnoses for both constituent genera Shiinoa Kabata, 1968 and Parashiinoa West, 1986, transfer Shiinoa rostrata Balaraman, Prabha & Pillai, 1984 to Parashiinoa as Parashiinoa rostrata (Balaraman, Prabha & Pillai, 1984) n. comb., and present keys to the females and males of both genera.

Introduction

Copepods are a diverse group of small crustaceans comprising about 14,690 species (Walter & Boxshall, 2023) including approximately 5,500 parasitic species (Bernot, Boxshall & Crandall, 2021). The copepod body is typically composed of a cephalosome comprising six body segments (i.e., somites) and a posterior trunk comprising nine somites plus the anal somite that represents the telson (Huys & Boxshall, 1991). Each somite of the cephalosome bears a pair of limbs (i.e., the antennules, antennae, mandibles, maxillules, maxillae, and maxillipeds). The trunk is composed of five anterior somites known as pedigers that each bear a pair of swimming legs, while the posterior four somites lack appendages and comprise the genital somite and three abdominal somites, the last of which is the anal somite that bears the caudal rami. There are deviations from this format, and the most extreme are exhibited by various parasitic copepods (Huys & Boxshall, 1991; Boxshall & Halsey, 2004; Bernot, Boxshall & Crandall, 2021).

One of the most unusual variations of copepod body organization is seen in members of the cyclopoid family Shiinoidae Cressey, 1975. The family includes nine species of copepods parasitizing marine teleosts and was only discovered in 1968 (Kabata, 1968). Very little is known about their biology, evolutionary relationships, internal anatomy, or development, but it is clear they have a unique body form and attachment mechanism. The body of shiinoids exhibits many cases of reduced segmentation and segmental fusions. For example, legs in the adults are not functional for swimming and most species completely lack legs 4 and 5. Most strikingly, the females possess a pair of large, elephant tusk-shaped antennae that are anteriorly directed and oppose a rostrum formed from an outgrowth of cuticle between the antennules, which can attain 30% of their body length (Figs. 1A–1C). Together the antennae and the rigid rostrum form a clasp that shiinoids use to attach to their teleost hosts, usually to gill-like tissue in the nostrils (i.e., nasal lamellae) or, rarely, to the skin. This is the main, and perhaps only, way that female shiinoids attach to their hosts given the relatively small size of all their other appendages. This attachment mechanism, with antennae that clamp against another body region, is not known in any other group of copepods.

Figure 1 Colorized microCT volume rendering of Shiinoa inauris.

Female copepod (purple), male (blue), host tissue (red), egg sacs of female not colorized (USNM 1615601). (A) Lateral view, male grasping onto female at base of rostrum. (B) Ventral view showing oral region. (C) Anterio-lateral view showing ridges enhancing rigidity of rostrum. Abbreviations: An, antenna; Lab, labrum; L1–3, legs 1–3; Ro, rostrum.

The present study was undertaken to better characterize shiinoid anatomy and their unique mode of attachment. To identify the muscles used for attachment in both female and male shiinoids, we examined specimens of Shiinoa inauris Cressey, 1975 in situ attached to host tissue using micro-computed tomography (microCT) and examined specimens of other species with light microscopy. The precise composition and homology of the different body regions throughout the family has been clouded by the reduced external expression of body segmentation and the extreme reduction of legs 4 and 5; in particular, the fates of the fifth pediger and the genital and abdominal somites have been uncertain. Here, the use of microCT to examine the trunk musculature has provided robust new evidence which resolves these uncertainties. Finally, as more taxa have been described, exceptions have been found to the diagnostic characters originally used to differentiate between Shiinoa and Parashiinoa, so we revise the diagnoses of both here and present keys to the females and males of both genera.

Materials and Methods

Specimens

For this study, we examined specimens of Shiinoa elagata Cressey, 1976, Shiinoa inauris, Shinnoa occlusa Kabata, 1968, Parashiinoa mackayi West, 1986, and Parashiinoa sp., including the types of S. inauris, S. occlusa, and P. mackayi from the Invertebrate Zoology Collections of the National Museum of Natural History, Smithsonian Institution and the Natural History Museum, London. Details of the specimens examined are as follows: three specimens of Shiinoa elagata (USNM 282624–282626); S. inauris holotype (USNM 142660), allotype (USNM 142661), three female and one male paratypes (USNM 142662), 19 females and eight males (USNM 229106–229116), and one female and one male (BMNH.1990 37-38); the holotype of S. occlusa (BMNH 1966.12.2.1) and two females and one male of S. occlusa (BMNH.1990 49-51); two female and one male paratypes of P. mackayi (BMNH 1986.1-3); two specimens of Parashiinoa sp. (USNM 266560). For morphological observation with light microscopy, specimens were temporarily mounted on cavity slides and cleared in lactic acid for at least 1 h prior to observation with an Olympus BX51 compound microscope. Fish taxonomy follows Fishbase (Froese & Pauly, 2023).

MicroCT

Three samples of nasal lamellae from species of Scomberomorus Lacepède, 1801 with attached specimens of Shiinoa inauris were examined with microCT at the National Museum of Natural History (NMNH). These comprised USNM 1615601 from Scomberomorus maculatus (Mitchill, 1815) and USNM 229112 from Scomberomorus sp., each consisting of a piece of nasal lamella with a single attached copepod, and USNM 229107, a large, nearly complete piece of nasal lamellar tissue from Scomberomorus maculatus with two females and two males attached to the females. These three pieces of tissue with attached copepods were prepared for microCT by being submerged in a single solution of 0.5% phosphotungstic acid (PTA) in 70% ethanol for 19 days. Specimens were mounted in 200 µl pipet tips filled with 0.5% low melt agarose in reverse osmosis purified water and sealed with paraffin wax. Specimens were scanned using the GE Phoenix v|tome|x M 240/180 kV Dual Tube microCT machine at the Micro Computed Tomography Imaging Center (mCTIC) at the Smithsonian Institution’s National Museum of Natural History (NMNH). Voxel sizes ranged from 2–4.5 µm. The raw CT data were reconstructed using GE Datos 2 and slices were exported using VG Studio Max 3.2. Additional scan settings are detailed in Table S1. Scans were visualized using 3DSlicer v4.13 (Kikinis, Pieper & Vosburgh, 2014) and Dragonfly v2022.2 (Dragonfly, 2022). 3DSlicer was used to visualize volume renderings by restricting the displayed grey values to omit background and mounting media, and male, female and host tissue were then colorized using Adobe Photoshop v25.3.1. Dragonfly was used to visualize slices with colorized female body and muscle tissue and to manually segment meshes highlighting the female body and muscles.

Results

MicroCT results

Female Shiinoa inauris attachment

Adult females attach to the nasal lamellae within the nostrils of their hosts (Fig. 2, Video S1). All host and locality records for S. inauris and the other species of Shiinoidae are given in Table 1. Females attach using a clasp-like mechanism wherein the pair of large, curved antennae that extend ventrally and anteriorly, are swung upwards to oppose an extremely elongated, rigid rostrum, formed as an outgrowth of the frontal margin of the dorsal cephalothoracic cuticle between the antennules (Figs. 1A–1C). In all observed specimens, the distal parts of both antennae were in direct contact with the rostrum, having punctured through a narrow fold of host lamellar tissue, leaving a single hole in the tissue through which the rostrum and both antennae pass (Fig. 1A). This hole was substantially larger than the cross-sectional area of the antennae and rostrum (Fig. 1B), suggesting that the host tissue had been further eroded by mechanical damage or perhaps from the feeding activities of the parasite.

Figure 2 Colorized microCT volume rendering of nasal lamellae of Scomberomorus maculatus with copepods attached.

Host tissue (red) with two female (purple) and two male (blue) specimens of Shiinoa inauris attached (USNM 229107). An asterisk (*) indicates atypical attachment location of male.

Table 1 Shiinoidae species hosts and localities.

Species	Host species	Host family	Locality	Reference	
Shiinoa elagata	Elegatus bipinnulata (as E. bipinnulatus)	Carangidae	Caroline Islands	Cressey (1976)	
	Elagatis bipinnulata (as Seriolichthys bipinnulatus)	Carangidae	Trivandrum	Balaraman, Prabha & Pillai (1984)	
	Elegatus sp.	Carangidae	Gulf of Thailand; Vanikoro Island	Cressey (1976)	
	Parastromateus niger	Carangidae	Trivandrum	Balaraman, Prabha & Pillai (1984)	
	Platax orbicularis	Ephippidae	Trivandrum	Balaraman, Prabha & Pillai (1984)	
	Sphyraena barracuda	Sphyraenidae	Belize	Cressey & Cressey (1986)	
Shiinoa inauris	Scomberomorus regalis	Scombridae	Cuba; Gulf of Venezuala; Surinam	Cressey (1975)	
	Scomberomorus brasiliensis	Scombridae	Surinam, Brazil	Cressey & Cressey (1980)	
	Scomberomorus maculatus	Scombridae	Texas; Florida (Key West, Placida); Argentina	Cressey (1975)	
Shiinoa japonica	Kyphosus vaigiensis	Kyphosidae	Tanabe Bay, Japan	Izawa (2009)	
Shiinoa occlusa	Scomberomorus commerson (as S. commersoni)	Scombridae	Green Island, Australia	Kabata (1968)	
	Acanthocybium solandri	Scombridae	Kapingarmarangi Atoll	Cressey & Cressey (1980)	
	Acanthocybium solandri	Scombridae	Trivandrum	Balaraman, Prabha & Pillai (1984)	
	Grammatorcynus bicarinatus	Scombridae	North Celebes	Cressey (1976)	
	Grammatorcynus bicarinatus	Scombridae	Solomon Islands; Palau; Caroline Islands	Cressey & Cressey (1980)	
	Gymnosarda unicolor	Scombridae	Solomon Islands	Cressey & Cressey (1980)	
	Scomberomorus commerson (as S. commersoni)	Scombridae	Arabian Sea	Cressey (1975)	
	Scomberomorus commerson	Scombridae	Mozambique; Pakistan; Gulf of Thailand; Solomon Islands; Philippines; Palau	Cressey & Cressey (1980)	
	Scomberomorus guttatus	Scombridae	China	Cressey & Cressey (1980)	
	Scomberomorus niphonius	Scombridae	Japan	Cressey & Cressey (1980)	
	Scomberomorus queenslandicus	Scombridae	Palau	Cressey & Cressey (1980)	
	Scomberomorus tritorr	Scombridae	Canary Islands	Cressey & Cressey (1980)	
Shiinoa prionura	Prionurus scalprum	Acanthuridae	Tanabe Bay, Japan	Izawa (2009)	
	Oplegnathus fasciatus	Oplegnathidae	Shima, Mie Prefecture, Japan	Izawa (2023)	
Parashiinoa bakeri	Haemulon sciurus	Haemulidae	Cuba	Cressey & Cressey (1986)	
	Aprion virescens	Lutjanidae	Western Indian Ocean	Cressey & Cressey (1986)	
	Carangoides malabaricus	Carangidae	Ethiopia	Cressey & Cressey (1986)	
	Caesio caerulaurea (as C. caerulaureus)	Caesionidae	Western Indian Ocean	Cressey & Cressey (1986)	
	Haemulon carbonarium (as H. carbonarum)	Haemulidae	Colon, Panama	Cressey & Cressey (1986)	
	Lutjanus kasmira (as L. kasimira)	Lutjanidae	Algoa Bay, South Africa	Cressey & Cressey (1986)	
	Macolor niger	Lutjanidae	Seychelles	Cressey & Cressey (1986)	
Parashiinoa cookeola	Cookeolus japonicus	Priacanthidae	Tanabe Bay, Japan	Izawa (2009)	
Parashiinoa mackayi	Pomadasys maculatus	Haemulidae	Cleveland Bay, Australia	West (1986)	
	Pomadasys argenteus	Haemulidae	Cleveland Bay, Australia	West (1986)	
Parashiinoa rostrata n. comb.	Priacanthus hamrur	Priacanthidae	Trivandrum	Balaraman, Prabha & Pillai (1984)	
Note:

Entries in bold indicate type records.

Analysis with microCT revealed details of the muscles involved in the clasping movements of the antennae (Figs. 3A–3D). No muscles are present in the rostrum, therefore the clasping action is generated by the antennae alone as they swing forwards and upwards to oppose the rigid, chitinized rostrum. The rigidity of the rostrum appears to be enhanced by a median ridge and pair of lateral ridges which extend from its base to about two thirds of its length (Fig. 1C). The clasping mechanism is generated by two sets of muscles: a set of large, extrinsic muscles originating dorsally in the anterior cephalothorax and inserting in different positions around the rim of the proximal segment of the antennae, and a pair of intrinsic muscles located within the mid-region of the antennae and operating across the articulation between antennal segments one and two.

Figure 3 MicroCT slices through female Shiinoa inauris antennal musculature.

(USNM 229112) (A) Axial cross section at the level of the base of the antennae showing course of extrinsic promotor muscles 1–3. (B) Axial cross section at the level of the base of the antennae showing course of extrinsic remotor 1 muscle. (C) Saggital section at the base of the antenna showing extrinsic antennal promotors 1 and 2 inserting anteriorly on proximal rim of antenna. (D) Saggital section at the base of the antenna showing extrinsic antennal remotor 1 inserting posteriorly on rim of antenna. (E) Saggital section at the middle region of the antenna showing intrinsic flexor and extensor muscles. (F) Sagittal section at the middle region of the antenna showing intrinsic extensor. Abbreviations: An, antenna; EPr1–3, extrinsic promotors 1–3; ERe1–2, extrinsic remotors 1–2; IE, intrinsic extensor; IF, intrinsic flexor; Ro, rostrum.

The extrinsic musculature comprises several muscle bundles which serve to swing the antennae forwards and up towards the underside of the rostrum, or backwards and down, away from the rostrum. These are the muscles responsible for the forward (promotor muscles) and backward (remotor muscles) motions of the whole antenna. They originate on the inner surface of the dorsal cephalothoracic shield and insert on the rim of the proximal antennal segment (Figs. 3A–3D). There are three promotor muscles: extrinsic promotor 1 (EPr1) originates along a ridge-like apodeme, an interiorly directed sagittal crest-like structure, that extends internally along the midline of the cephalothoracic shield (Figs. 3A, 3B). EPr1 has a broad origin and tapers markedly towards its anteromedial insertion on the rim of the antenna (Fig. 4A). Extrinsic promotor 2 (EPr2) also originates close to the midline (Fig. 4A) and passes transversely downwards to insert anterolaterally on the rim of the antenna. Extrinsic promotor 3 (EPr3) originates directly dorsal to the foramen at the base of the antenna (Fig. 4A) and passes ventrally to insert anterolaterally on the rim of the antenna. The two extrinsic remotor muscles (ERe1, ERe2) (Fig. 4A) are longer and more slender than the promotors. The extrinsic remotors originate on the dorsal wall of the cephalothorax anterior to the base of the limb and pass obliquely posteroventrally to insert posteriorly on the expanded rim of the proximal segment of the antenna. Together the three promotors have a much larger cross-sectional area than the opposing remotors.

Figure 4 Schematic illustration of antennal musculature of female Shiinoa inauris.

(A) Dorsal view looking down into proximal opening (i.e., foramen) of antenna, showing extrinsic muscles, designated as promotors 1–3 and remotors 1–2 inserting around rim of proximal antennal segment. (B) Lateral view of whole antenna showing opposing pair of intrinsic muscles spanning first to second segment articulation, designated as intrinsic flexor and extensor. Schematic illustrations based on light microscopy examination. Abbreviations: An, antenna; EPr1–3, extrinsic promotors 1–3; ERe1–2, extrinsic remotors 1–2; IE, intrinsic extensor; IF, intrinsic flexor.

The antenna comprises a long curved first segment and a slightly shorter, curved second segment terminating in a blunt, digitiform tip (Fig. 4B). The second segment is compound (comprising at least two ancestral endopodal segments) and there is an internal cuticular structure with a bifid tip which may represent a modification of the original articulation between the component segments. The intrinsic musculature comprises an opposing muscle pair: both muscles originate within the first antennal segment and insert inside the proximal rim of the second segment. The intrinsic flexor (IF) (Figs. 3E, 4B) is short and tapers markedly from its broad distal origin in the first segment towards its insertion on the rim of the second. Contraction of this flexor muscle pulls the tip of the antenna up and into contact with the underside of the rostrum. The opposing intrinsic extensor (IE) (Figs. 3F, 4B) is a longer, more slender muscle originating about at mid-length of the long curved first antennal segment. It inserts inside the rim of the second antennal segment and contracts to pull down on the tip of the antenna so that it disengages from contact with the rostrum. These intrinsic muscles move the second segment of the antenna relative to the proximal first (Fig. 4B).

Male attachment

Shiinoid males are apparently always found attached to females. There are no reports of any males attached directly to the host, but males are less commonly found, and attachment location has not been documented for all species. In Shiinoa, males are usually found attached to females at the base of the rostrum just behind the female antennae while in Parashiinoa, males attach more posteriorly (Fig. 5, Table 2). Males lack the elongate rostrum that is present in the female and attach to the female using their large antennae that extend latero-ventrally around the body of the female. On one female of S. inauris two males were attached, with one male grasping the female behind the base of the rostrum and the other attaching at the boundary between pedigers 2 and 3 (Figs. 2, 5A, Video S1).

Figure 5 Schematic illustration of body plans of shiinoid females and male attachment locations.

Major body regions labeled: Lab, labrum; P2/P3, pediger 2 and 3 boundary; P5/GS, pediger 5 and genital segment boundary; arrowhead indicates male attachment location; asterisk indicates atypical location of second male specimen of Shiinoa inauris in Fig. 2 (USNM 229107). (A) Shiinoa inauris (adapted from Cressey, 1975). (B) Shiinoa occlusa (adapted from Balaraman, Prabha & Pillai, 1984). (C) Parashiinoa mackayi (adapted from West, 1986). (D) Parashiinoa bakeri (adapted from Cressey & Cressey, 1986).

Table 2 Shiinoidae species male size and attachment location.

Species	Male:female body length	Male attachment location	
Shiinoa elagata	0.74 (Balaraman, Prabha & Pillai, 1984)	Unknown	
Shiinoa inauris	0.51 (Cressey, 1975)	Behind antenna (Cressey, 1975)†	
Shiinoa japonica	0.48 (Izawa, 2009)	Unknown	
Shiinoa occlusa	<0.61* (Cressey, 1975)	Behind antenna (Cressey, 1975)	
Shiinoa prionura	0.60 (Izawa, 2023)	Unknown	
Parashiinoa bakeri	0.24 (Cressey & Cressey, 1986)	Genital segment (Cressey & Cressey, 1986)	
Parashiinoa cookeola	>0.1** (Izawa, 2009)	P5/genital segment boundary (Izawa, 2009)	
Parashiinoa mackayi	0.43 (West, 1986)	P2/P3 boundary (West, 1986)	
Parashiinoa rostrata n. comb.	0.22 (Balaraman, Prabha & Pillai, 1984)	P5/genital segment boundary (Balaraman, Prabha & Pillai, 1984)	
Notes:

* Female body length based on immature specimen.

** Only male copepodid II known.

† In this study a second male on the same female was found attached at P2/P3 boundary.

The antenna of a male shiinoid is a short, grasping appendage with a long proximal (protopodal) segment and a shorter distal part derived from the endopodal segments. The distal part is typically two-segmented and carries between 2 and 4 stout, claw-like elements plus several smaller setae. The proximal segment extends down around the lateral surface of the female and the distal part flexes medially to grasp onto the female (Figs. 1A, 1C).

The antennal musculature of the male is similar to that of the female. The extrinsic muscles originate on the dorsal cephalothoracic shield (Figs. 6A, 6B) and pass into the base of the antenna, inserting around the rim of the proximal antennal segment. The extrinsic musculature comprises adductor muscles that swing the entire antenna medially, to grasp around the female body, and abductor muscles which swing the limb laterally away from the female, thereby releasing the grip of the male on the female. The sexual dimorphism in antennal motions, which are primarily moved forwards and backwards in the female compared to moving in towards the midline and out again in the male, is reflected in the antennal musculature primarily in the position of the muscle insertions on the proximal rim of the antenna. The male lacks the ridge-like apodeme extending internally along the midline of the cephalothoracic shield which is present in the female. The adductors and abductors of the male are homologous to the promotors and remotors of the female, respectively, with the terminology changing to reflect the differing motions of the antennae in the two sexes.

Figure 6 MicroCT slices showing male Shiinoa inauris antennal musculature.

Male copepod (blue), female (purple), muscles (orange) (USNM 229112). (A) Long intrinsic extensor muscle in proximal segment of antenna. (B) Dorsal origins of extrinsic muscles to antenna, and short intrinsic flexor muscle within antenna. Abbreviations: IE, intrinsic extensor; IF, intrinsic flexor.

The intrinsic musculature of the male is the same as in the female. An opposing muscle pair operates across the articulation between segments 1 and 2 of the antenna. The shorter, conical intrinsic flexor muscle (IF) has a broad origin distally in the first antennal segment (Fig. 6B), passes obliquely across the limb and inserts on the proximal rim of the second segment. The longer intrinsic extensor muscle (IE) (Fig. 6A) originates near the middle of the first segment and inserts on the proximal rim of the second segment. The flexor muscle acts to bring the distal part of the antenna closer to the midline and bring the distal array of claws into contact with the surface of the female. The extensor muscle straightens the antenna, pulling the distal part away from the midline and withdrawing the claws from contact with the female.

Body segmentation and trunk musculature

The main body musculature of S. inauris comprises paired dorsal (dlm) and ventral (vlm) longitudinal trunk muscle bundles (Fig. 7A). Both dlm and vlm originate in the cephalothorax and pass longitudinally back along the body on either side of the midline. The dlm have broad dorsal origins in the antennal region of the anteriorly-extended cephalothorax of S. inauris. They pass posteriorly through the cephalothorax to an intermediate attachment at the plane of the articulation between the cephalothorax (incorporating pediger 1) and pediger 2, and then pass posteriorly making intermediate tendinous attachments at the planes of the pediger 2/pediger 3, pediger 3/pediger 4, and pediger 4/pediger 5 articulations (Figs. 7A, 7B), before inserting on the anterior rim of the genital segment. The antero-posterior passage of the dlm appears interrupted in the genital segment by the musculature associated with the genital apparatus, but the dlm continues again through the abdomen to finally insert at midlength of the posterior-most abdominal segment (which bears the caudal rami).

Figure 7 MicroCT segmented meshes of female Shiinoa inauris body and trunk musculature.

(A) Whole copepod, lateral view (USNM 1615601). (B) Posterior body from pediger 4 to caudal rami, dorsal view (USNM 229112). Abbreviations: Abd1–2, abdominal segments 1–2; Abd3/4, compound distal abdominal segment; An, antenna; CR, caudal rami; GS, genital segment; Lab, labrum; L1–3, legs 1–3; P4–5, pedigers 4–5; Ro, rostrum.

The ventral longitudinal trunk muscles follow a similar course: originating ventrally in the cephalothorax and passing posteriorly through the cephalothorax and all post-cephalothoracic trunk segments before finally inserting at midlength of the posterior-most (third) abdominal segment. Intermediate attachments via short tendinous inserts are present at the planes of the cephalothorax/pediger 2, pediger 2/pediger 3, pediger3/pediger 4, pediger 4/pediger 5, pediger5/genital segment, genital segment/abdominal segment 1, and abdominal segment 1/abdominal segment 2 articulations (Figs. 7A, 7B).

Taxonomic results

The genus Parashiinoa was differentiated from Shiinoa by West (1986) based on three character states: the rami of both legs 1 and 2 are two-segmented rather than three-segmented, the male attachment position is at the genital region rather than near the head of the female behind the antennae, and the female antennae are held laterally rather than vertically. With the discovery of new species attributed to both genera, these characters all require reassessment and new diagnoses for both genera are provided here.

Shiinoa Kabata, 1968

Diagnosis: Female body slender and cylindrical or dorso-ventrally flattened. Anterior part of cephalothorax bearing antennules and antennae extended anteriorly; frontal margin of antennal extension produced into elongate, rigid rostrum. Labrum, mouthparts and leg 1 located in posterior part of cephalothorax. Post-cephalothoracic body comprising second to fifth pedigers, genital segment bearing paired genital apertures, and free abdomen comprising one to three free segments. Second pediger typically distinct, forming narrow “waist” in some species. Third to fifth pedigers separate or, typically, fused to form posterior trunk region. Antennule six- or seven-segmented. Antenna two-segmented with compound digitiform terminal segment (with subdivisions in S. elagata, S. japonica, and S. prionura). Legs 1 and 2 biramous with three-segmented exopods and typically with three-segmented endopods, except in S. inauris (two-segmented). Leg 3 comprising two-segmented protopod with outer seta on basis and one-segmented exopod bearing two spines and one outer seta.

Male body slender, tapering posteriorly; comprising cephalosome, first to fifth pedigers, genital segment and four abdominal segments. Rostrum small, extending posteroventrally from frontal margin of cephalic shield. Antennule seven-segmented or six-segmented with first two segments forming compound segment; posterodistal corner of first and second segments of six-segmented forms drawn out into long spinous process. Antenna three-segmented with distal segment bearing 3 or 4 claws. Legs 1 and 2 biramous, with three-segmented rami, except in S. inauris (two-segmented).

Type species: Shiinoa occlusa Kabata, 1968

Other species: Shiinoa elagata Cressey, 1976, Shiinoa inauris, Cressey, 1975, Shiinoa japonica Izawa, 2009, Shiinoa prionura Izawa, 2009

Remarks

The antenna in species of Shiinoa has an elongate proximal segment and a distal compound segment that each make up about half the length of the appendage. Shiinoa occlusa and S. inauris resemble each other in their elongate cylindrical body form and possession of antennae with only two distinct segments. In the other species of Shiinoa, the distal compound segment has irregular subdivisions. Shiinoa elagata has a knuckle-like structure that does not resemble a true segment since it is not a complete subdivision while S. prionura and S. japonica as illustrated by Izawa (2009) have multiple subdivisions distally. Izawa (2009) described the antenna of both species as five-segmented including an additional small segment at the base of the limb. Given that we did not observe any segmental division in this region in S. inauris and that there was no evidence from the musculature to indicate a separate basal segment, we suspect this is not a true segment.

Shiinoa prionura is similar to S. elagata and S. japonica. These three species have very similar setation patterns on the swimming legs in the female. They are the only species of shiinoids that possess an outer spine on the second exopodal segment of legs 1 and 2, and an inner seta on the first endopodal segment of leg 2. Shiinoa prionura and S. elagata also possess an inner seta on the second exopodal segment of leg 2 but this is absent in S. japonica. Izawa (2009) distinguished between the females of S. prionura and S. elagata by the presence in the former of vestiges of leg 4, represented as a pair of rounded lobes located on the ventral body surface just posterior to leg 3, and by the more slender body of S. prionura. Izawa (2023) redescribed the female of S. prionura and described the male for the first time. The male of S. prionura shows no vestige of leg 4, even though this was described as present in the female. The robustness of these differences between S. prionura and S. elagata should be tested as more material becomes available.

Parashiinoa West, 1986

Diagnosis: Female body dorso-ventrally flattened. Anterior part of cephalothorax bearing antennules and antennae extended anteriorly; frontal margin of antennal extension produced into typically elongate rostrum (rostrum short in Parashiinoa mackayi). Labrum, mouthparts and leg 1 located in posterior part of cephalothorax. Cephalothorax typically separated from posterior trunk region by narrow “waist” formed by second pediger. Posterior trunk region incorporating third to fifth pedigers. Genital segment bearing paired genital apertures. Free abdomen one- or two-segmented. Antenna apparently three-segmented with third segment typically forming terminal blade (but irregularly lobate in P. mackayi). Legs 1 and 2 biramous with both rami two-segmented. Leg 3 comprising two-segmented protopod with outer seta on basis and one-segmented exopod bearing at least two spines, or with protopod incorporated into body segment and exopod reduced to bisetose lobe.

Male body slender, tapering posteriorly; comprising cephalosome, first to fifth pedigers, genital segment, and two or four abdominal segments. Rostrum small, extending posteroventrally from frontal margin of cephalic shield. Antennule seven-segmented or six-segmented with all segments short. Antenna three-segmented with distal segment bearing two or three claws. Legs 1 and 2 biramous with two-segmented rami.

Type species: Parashiinoa mackayi West, 1986

Other species: Parashiinoa bakeri (Cressey & Cressey, 1986), Parashiinoa cookeola Izawa, 2009, Parashiinoa rostrata (Balaraman, Prabha & Pillai, 1984) new combination

Remarks

Shiinoa rostrata Balaraman, Prabha & Pillai, 1984 shares important character states with Parashiinoa as newly diagnosed here. In particular, it possesses two-segmented endopods on legs 1 and 2 in the female, the antenna of the female is three-segmented with a blade-like terminal segment and the endopods of legs 1 and 2 are armed with a single well developed apical spine in females. On this basis, we here transfer Shiinoa rostrata to Parashiinoa as P. rostrata (Balaraman, Prabha & Pillai, 1984) n. comb.

Izawa (2009) established P. cookeola based on material from a priacanthid host (Cookeolus japonicus) but made no comparisons with P. rostrata, the only other shiinoid found on a priacanthid host (Priacanthus hamrur) (Table 2). These species are very similar and are separated in the key below only by the presence/absence of the slender outer seta on the free exopodal segment of leg 3. This seta is present in the great majority of shiinoids and is present in P. cookeola, but was not figured by Balaraman, Prabha & Pillai (1984) in their description of P. rostrata. As more material of the latter species becomes available, the absence of this small seta should be verified in order to confirm the validity of the P. cookeola as distinct from P. rostrata.

Key to species of Shiinoidae (females)

1. Legs 1 and 2 biramous with both rami three-segmented2

Legs 1 and 2 biramous with two-segmented exopod and two- or three-segmented endopod5

2. Body slender, not dorsoventrally flattened, with paired postero-lateral processes on fifth pedigerS. occlusa

Body dorsoventrally flattened without conspicuous paired postero-lateral processes3

3. Legs 1 and 2 lacking inner seta on first endopodal segment; second exopodal segment of leg 2 without inner setaS. japonica

Legs 1 and 2 armed with inner seta on first endopodal segment; second exopodal segment of leg 2 with inner seta4

4. Lobate vestige of leg 4 present posterior to leg 3S. prionura

Leg 4 absentS. elagata

5. Legs 1 and 2 with three-segmented endopods and two-segmented exopodsS. inauris

Legs 1 and 2 with both rami two-segmented6

6. Rostrum elongate; legs 1 and 2 with distal endopodal segment bearing single apical spine7

Rostrum short; legs 1 and 2 armed with two spines plus setal vestigesP. mackayi

7. Distal exopodal segment of leg 1 armed with total of five spines along outer and distal margins8

Distal exopodal segment of leg 1 armed with total of six elements (five outer and distal spines plus one inner seta)P. bakeri

8. Distal segment of leg 3 armed with two spines plus one outer setaP. cookeola

Distal segment of leg 3 armed with two spines onlyP. rostrata

Key to species of Shiinoidae (males)

1. Legs 1 and 2 biramous with three-segmented rami2

Legs 1 and 2 biramous with two-segmented rami5

2. Antennule six-segmented with long distal process posteriorly on segment 2; second exopodal segment of leg 2 bearing outer margin spine3

Antennule seven-segmented; second exopodal segment of leg 2 lacking outer margin spineS. occlusa

3. Second exopodal segment of leg 2 with inner seta4

Second exopodal segment of leg 2 lacking inner setaS. japonica

4. Distal exopodal segment of leg 2 armed with three spines and two setaeS. elagata

Distal exopodal segment of leg 2 armed with three spines and three setaeS. prionura

5. Abdomen four-segmented6

Abdomen two-segmented7

6. Distal exopodal segment of leg 2 armed with two robust claws and two setaeP. mackayi

Distal exopodal segment of leg 2 armed with one large claw (longer than ramus) plus two outer spines and three inner setaeS. inauris

7. Antennule six-segmentedP. bakeri

Antennule seven-segmentedP. rostrata

Discussion

Attachment mechanism

Members of Shiinoidae exhibit an attachment mechanism unique among copepods, which varies between genera and sexes. Despite comprising only nine species, the body form of females is heterogeneous (Fig. 5). There are two distinct body types: a slender, more-or-less cylindrical form, as found in Shiinoa occlusa and S. inauris (Figs. 5A, 5B), and a dorsoventrally flattened form found in the other species of Shiinoa and in all species of Parashiinoa (e.g., Figs. 5C, 5D). Parashiinoa mackayi is arguably a third form because it is the only species that lacks a rostrum. Females of all species of Shiinoa attach with antennae that extend anterio-ventrally, nearly parallel to each other, and oppose their elongate rostrum (e.g., Figs. 5A, 5B). Parashiinoa mackayi and males of both genera lack an elongate rostrum and attach with laterally opposed antennae whose distal ends form a clamp against each other or against the body of the female (Figs. 5C, 1B, 1C). Females of all other species of Parashiinoa attach in a somewhat intermediate fashion in which the antennae are ventro-laterally opposed forming a clamp against the rostrum (Fig. 5D). We hypothesize the attachment mechanism used by females of P. mackayi is the ancestral mechanism because the laterally opposing antennae are not dependent on a rostrum. Given that no other copepods possess such a highly elongated rostrum, it seems likely that evolution of this attachment mechanism proceeded from a form lacking a rostrum to a more derived, elongated rostral form. However, the evolutionary history is unclear because the phylogenetic relationships of shiinoids remains unknown.

No other parasitic copepods utilize an attachment mechanism in which the antennae form a clasp against an elongate cuticular outgrowth originating from the main body. While the antennae of parasitic copepods are commonly modified for attachment, they are typically modified into hook or claw-like structures that individually grasp host tissue. Some genera of Ergasilidae resemble the Shiinoidae in their possession of highly elongate antennae. For example, in many ergasilid genera such as Acusicola Cressey, 1970, the antennae oppose each other, with left and right antennae interlocking at their tips (El-Rashidy & Boxshall, 1999), encircling a gill filament in a hugging-like fashion. However, ergasilids do not possess an elongate rostrum and the antennae do not clasp against any element of the cephalothorax.

Other pancrustaceans have evolved rostrum-like structures formed from outgrowths of segments of the head. Among copepods, the only such structure that we are aware of that is potentially used for attachment is found in a cluster of nostril-inhabiting parasitic copepods belonging to Bomolochidae and comprising the genera Acanthocolax Vervoort, 1969, Ceratocolax Vervoort, 1965, Tegobomolochus Izawa, 1976, and perhaps Unicolax Cressey & Cressey, 1980. These taxa possess a pair of protuberances on the anterodorsal surface of the cephalothorax that appear to form pincer-like structures with elements on the first segment of the antennule (Huys et al., 2012, Bernot & Boxshall, 2019). But unlike shiinoids, the structures in bomolochids are paired, much smaller (3% of body length vs >20%), opposed by small structures on the antennules rather than elongate antennae, and have yet to be documented clamping host tissue. Outside of Copepoda, other pancrustacean possess rostrums formed from head segments, but they are used for different functions. Decapod shrimp, hyperiid amphipods, lophogastrids, and zoea larvae possess rostrums that are involved in predator defense and stabilization (Smith & Jensen, 2015). Several insects possess similar structures, such as the frontal horn in rhinocerous beetles used for male combat and head processes in lanternflies (Fulgoridae) whose function is poorly known but does not appear to be involved with grasping of any kind (Urban & Cryan, 2009). The most structurally and functionally similar grasper is probably that of the Cretaceous hell ants (Haidomyrmecinae). Like shiinoids, the rostrum is formed by an outgrowth of cuticle between the first pair of appendages, but in hell ants it is opposed by elongate, anteriorly directed mandibles rather than antennae. Recently discovered specimens in amber show that the rostrum of the hell ant Ceratomyrmex ellenbergeri Perrichot, Wang, and Engel, 2016 was used to form a comparable clasp for capturing prey (Barden, Perrichot & Wang, 2020; Perrichot, Wang & Barden, 2020), a fascinating example of convergent evolution in a group of distantly related pancrustaceans. It would be interesting to explore the musculature and developmental processes involved in the formation of the various pancrustacean rostrums.

Body segmentation and trunk musculature

In free living copepods, such as calanoids and misophrioids, the main body musculature comprises paired dorsal (dlm) and ventral (vlm) longitudinal trunk muscle bundles (Boxshall, 1982, 1985). These originate in the cephalothorax and pass along the body forming intermediate tendinous attachments at each segmental boundary before finally inserting on the anterior rim of the fourth abdominal (i.e., anal) segment. The basic body musculature pattern is very similar in shiinoids, although the locations of the origins of both dlm and vlm have been impacted by the anterior-extension of the cephalothorax. The origin of the dlm is more anteriorly located in shiinoids and lies close to the base of the rostrum, presumably shifted forward with the evolutionary elongation of the cephalothorax (Fig. 7). The vlm originate on apodemes in the oral region (associated with the maxillules and maxillae) in free living copepod orders (Boxshall, 1982, 1985) but these could not be resolved with microCT here. The vlm are attached in the oral region but there appear to be muscles extending anteriorly from these attachments towards origins on the ventral body wall in the extended antennal part of the cephalothorax. These may have resulted from an anterior shift in the origin of the vlm or the anterior part may represent other muscles that have been conscripted to effectively form an anterior extension of the vlm.

Establishing the precise composition and homology of the different body regions throughout the Shiinoidae has been hindered by the loss of legs 4 and 5 and reduction in the outward expression of body segmentation patterns. In particular, the fates of the fifth pediger and the genital and abdominal somites have remained uncertain. The use of microCT to visualize the body trunk musculature provided new evidence that resolves these uncertainties. The pattern of intermediate attachments in both dlm and vlm confirms that pediger 5 is fused to pediger 4 in females, rather than incorporated into the genital segment to form a genital complex. Given the tagmosis of the dorso-ventrally flattened forms such as the species of Parashiinoa, it is likely that pediger 5 forms part of the posterior trunk region and that the genital segment remains free in all shiinoids (Fig. 5).

The other uncertainty regarding body segmentation patterns in Shiinoidae relates to abdominal segmentation in females. The ancestral state in Copepoda is the presence of a distinct genital segment followed by four free abdominal segments (Huys & Boxshall, 1991), the last of which is traditionally referred to as the anal segment. Shiinoa occlusa and S. inauris display only three segments posterior to the genital segment (Figs. 5A, 5B), and some other female shiinoids have even fewer (Fig. 5D). The pattern of longitudinal trunk muscles in S. inauris indicates that the posteriomost division of the body (that carries the caudal rami) is probably compound, comprising the third and fourth (= anal) abdominal segments which have not separated during development. The dlm and vlm pass into this compound segment inserting about at mid-length, marking the plane of the ancestral segmental boundary (Fig. 7B). In free-living copepods, the longitudinal trunk muscles insert on the anterior (proximal) rim of the anal (fourth abdominal) segment (Boxshall, 1982, 1985). The mid-segment location of the insertion in S. inauris indicates that this is likely a compound segment comprising the third and fourth abdominal segments, and therefore also confirms that the genital segment is a single segment rather than the typical genital double-somite formed by the fusion of the first abdominal and genital segments found in most podoplean copepods (Huys & Boxshall, 1991).

Systematics

Here we revised the generic diagnoses of Shiinoa and Parashiinoa given that several new species have been described and exhibit variation relative to the original diagnoses of the genera. In particular, our revision: (1) incorporates the two-segmented state of the endopod of legs 1 and 2 in S. inauris, which was previously regarded by West (1986) as diagnostic of Parashiinoa, (2) allows for heterogeneity within Parashiinoa in orientation of the antennae, (3) includes new differences in antennal segmentation between Shiinoa and Parashiinoa, and (4) provides numerous diagnostic features relating to the males and to the segmentation and setation of the other limbs of the females. West (1986) also differentiated Parashiinoa from Shiinoa by the male attachment site, noting that Parashiinoa males attached more posteriorly, near to the genital complex, rather than just behind the antennae. Until now, this pattern has helped distinguish the four species of Parashiinoa and the two species of Shiinoa for which male attachment location has been documented. However, in this study we found a female of S. inauris with two males attached to it, and while one male was attached to the rostral area the other was attached at the boundary between pediger two and three, just anterior to the genital segment. This may be because the rostral area was already occupied when the second male attached, but it demonstrates that the male attachment site can be variable.

The phylogenetic relationships of the Shiinoidae to the 102 other families of Cyclopoida (Walter & Boxshall, 2023) has remained elusive because of the unique morphology of shiinoids and the lack of molecular data for this family and most other parasitic cyclopoid families (Khodami et al., 2019; Bernot, Boxshall & Crandall, 2021). Kabata (1968) hypothesized Shiinoidae is closely related to Chondracanthidae based on morphology of the mandible but Ho (1971) disagreed. Cressey (1975) hypothesized Shiinoidae is most closely related to Ergasilidae based on morphology of several cephalic appendages and their means of host attachment using elongate antennae. Boxshall & Halsey (2004) hypothesized that Shiinoidae is related to Philichthyidae together in a family group that also includes Chondracanthidae and Lernaeosoleidae based on the morphology of the mandible. Huys et al. (2006) noted a similarity in the morphology of the mandible of Shiinoidae, Anthessiidae, and Myicolidae. Most recently, Izawa (2009) hypothesized a closer relationship between Shiinoidae and Philichthyidae than to Chondracanthidae based on highly reduced or absent maxillipeds in both sexes of shiinoids and philichthyids, and on morphological similarities in the males. Still, the phylogenetic relationships of Shiinoidae remain poorly resolved. Although we were unable to sequence specimens in this study, we hypothesize a relationship of Shiinoidae with Philichthyidae and Chondracanthidae is more likely than Ergasilidae based not only on the morphological similarities described by Boxshall & Halsey (2004) and Izawa (2009), but also on the mating system of these families. In the Chondracanthidae, Philichthyidae, and Shiinoidae a relatively small male attaches to the female for a prolonged period of time while the female is attached to a host. This is markedly different from the mating system in Ergasilidae where males and females mate briefly in the plankton and only females continue on to parasitize a host (Alston, Boxshall & Lewis, 1996). It would be particularly exciting to apply molecular sequence data to better resolve the phylogenetic relationships of Shiinoidae, which have remained a mystery since they were first described by Kabata (1968).

Conclusions

Females of the family Shiinoidae use a unique attachment mechanism in which large antennae clamp against a rostrum. Using microCT, we characterized the shiinoid musculature and show that this attachment mechanism is driven by four main muscle bundles of the antennae: three primary muscle bundles extend from the inner dorsal surface of the body into the proximal segment of the antenna (extrinsic promotors 1–3) to swing the entire limb forward and upward towards the rostrum while a single muscle bundle that extends between the first and second antennal segments (intrinsic flexor) pulls the distal segment up towards the rostrum. Opposing muscles serve to swing the appendage down and away from the rostrum (extrinsic remotors 1–2 and intrinsic extensor). MicroCT also revealed the pattern of longitudinal muscles extending along the main body, which enabled us to show that pediger 5 is fused to pediger 4 in females and that the genital segment is a single segment rather than the genital double-somite found in most podoplean copepods. This is an example of how microCT can help us understand attachment and body remodeling in extremely modified parasitic copepods. Several mysteries remain. The copepodid stages of female shiinoids have never been documented so the developmental progression of the rostrum and antennae are unknown. Given the highly reduced appendages of shiinoids and the fact that all specimens we observed had the rostrum and antennae passing through a single hole in tissue of their hosts, we are left wondering how initial settlement and attachment proceeds. The phylogenetic relationships among species of shiinoids are also unknown, as is their relationship to the rest of Copepoda, and therefore the evolution of their unique morphology and unusual attachment mechanism remains poorly understood.

Supplemental Information

Supplemental Information 1 Rotating video of microCT meshes of copepods and host tissue depicted in Figure 2 (USNM 229107).

Supplemental Information 2 MicroCT scanning details.

We thank William E. Moser and the collections management staff of the Smithsonian’s National Museum of Natural History, Department of Invertebrate Zoology Collections for their assistance with specimen and data management and Scott Whittaker, director of the Smithsonian’s National Museum of Natural History Scientific Imaging facility and mCTIC.

Additional Information and Declarations

Competing Interests

Author Contributions

Data Availability

The authors declare that they have no competing interests.

James P. Bernot conceived and designed the experiments, performed the experiments, analyzed the data, prepared figures and/or tables, authored or reviewed drafts of the article, and approved the final draft.

Geoffrey A. Boxshall conceived and designed the experiments, performed the experiments, analyzed the data, prepared figures and/or tables, authored or reviewed drafts of the article, and approved the final draft.

Freya E. Goetz conceived and designed the experiments, performed the experiments, analyzed the data, prepared figures and/or tables, authored or reviewed drafts of the article, and approved the final draft.

Anna J. Phillips conceived and designed the experiments, analyzed the data, authored or reviewed drafts of the article, and approved the final draft.

The following information was supplied regarding data availability:

All microCT scans and meshes are available on MorphoSource:

- https://doi.org/10.17602/M2/M562678 (USNM 229107),

- https://doi.org/10.17602/M2/M562672 (USNM 229112),

- https://doi.org/10.17602/M2/M575134 (USNM 1615601).

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
