# Peer review of "MicroCT illuminates the unique morphology of Shiinoidae (Copepoda: Cyclopoida), an unusual group of fish parasites"

_PeerJ, doi:10.7717/peerj.16966_

## Round 0.1 · original submission · Minor Revisions

Dear Authors

The manuscript received three comprehensive reviews. The reviewers have expertise in the method and Crustacea morphology respectively.

Their common request is for more detail about the method and to contextualize the importance of the study for non-copepod, non-parasitologists.

Various other minor comments are provided.

Will you please attend to all comments and provide a list of the comments and your responses.

·

Basic reporting

The paper “Revisiting the unique morphology of Shiinoidae (Copepoda: Cyclopoida) with microCT” (#91889) is a cool piece of work and should clearly be considered for publication in PeerJ. I have only few aspects that in my view still need improvement. Most of these stem from a single fact: PeerJ is not a copepod specialists journal. I have listed specific points further below that, as soon as improved, will increase the readability for the non-specialist and also make the story more attractive to a wider audience. Otherwise, nice piece of work!

Kind regards,
Joachim Haug, LMU Munich

Experimental design

Images are pretty good. Yet without a section how these were processed I am a bit lost, so the figure captions need more info. I, for example, think that Fig. 1 is a mixture of a volume render (the egg sacks?) and surface models (most of the rest), but that is a guess. If most of the body is indeed a surface render, it would be good to in addition show a volume render. Surface models are basically 3D camera lucida drawings and therefore it would be good to see the non-interpreted versions. The labels in the different figures are of quite different styles, it would be easier for the reader if that would be more uniform. Also some of the 3D reconstructions would benefit from more labels.

Validity of the findings

see "Experimental design"

Additional comments

The title is not very informative. This is not a specialists journal. I suggest to make it less taxonomic (especially without the brackets) and add the aspect of “fish parasite”, this will interest more people, maybe something like: “MicroCT reveals the unique morphology of Shiinoidae, copepodan fish parasites”

General note on use of taxonomy: Linnean ranks are arbitrary constructs. Families are cut down to subfamilies or raised to superfamilies with almost every second new paper coming out (overexaggeration). As this is not a taxonomic paper, I suggest to simply leave out all references to ranks and just say “Shiinoidae” instead of family. There is still the neutral expression “group” if necessary.

line 19 there is no “plan” in biology. This word is always a point for attack from creationists. A neutral term for “body plan” is “body organisation”.

line 21 clasper?

lines 33, 34 same comments as above

line 36, legs 4 and 5. Again not an expert journal. I have been working with different crustaceans for quite some time, I also can only guess that they mean trunk legs (right?), or is it post-maxillary legs? There is necessity for some more background info.

line 37 again I consider myself relatively knowledgable, but I have never heard of the term pediger. I suggest to avoid it as it can be simply substituted by the term “leg-bearing segment” (although as before it needs a short frame before what exactly is counted here).

line 87 I see no section how the resulting scans were processed. That would be quite cruzial for some aspects.

line 130 Usually segment is used for body parts, “segments” of appendages are either articles or elements.

line 361 Philosophically speaking phylogenetic unit (and therefore also taxonomic ones) have representatives, not members (for further discussion on this topic I suggest the authors to talk to their inhouse colleague Ronny Jenner)

lines 359ff. I fully agree that this attachment is rather unusual. For some further reaching discussion I suggest the authors to look at some phenomena beyond parasitism. For example there are other weird “scissoring-devices” (e.g. Bai, M., Beutel, R. G., Zhang, W., Wang, S., Hörnig, M., Gröhn, C., ... & Wipfler, B. (2018). A new Cretaceous insect with a unique cephalo-thoracic scissor device. Current Biology, 28(3), 438-443). Concerning the aspect of the rostrum it may be interesting to mention the “singing” of spiny lobsters in which the antennae are used against the rostrum. Another interesting similarity can be seen in Cretaceous hell ants in which the upward-moving mandibles grasp against a protrusion from the head.

The black value in Fig. 2 seems not to be black.

For the non-expert reader it would be really good to show a photo of one of the parasites attached to a fish.

Reviewer 2 ·

Basic reporting

This manuscript describes the morphology of shinoids using the micro-CT technology. Specifically, the authors focused on the attachment mechanism of these parasitic copepods and they also provided a taxonomic key related to the genera Shiinoa and Parashiinoa. They also provide a review of the known species of Shiinoidae accompanied by their host and locality data.
The manuscript is well written and adds important information related to the morphology of the shinoids filling a gap due to limited studies.
However, there are some minor issues that need to be clarified and they are listed below in details.
In the introduction section it would be better to add details about the micro-CT technology especially for the readers who are not familiar with this technology. For example you could refer to the micro-CT advantages. This tool has been used for several taxonomical studies so you could add some examples of its use in taxonomy (e.g. https://doi.org/10.2108/zs210063, https://doi.org/10.1371/journal.pone.0135243).
Information about shiinoids ecology and life cycle is missing. This fact may be related to the lack of studies in this topic as you also mentioned in the conclusions section, however some details for the parasitic copepods in general could be mentioned.

Line 23: please add the parent name in Shiinoa inauris Cressey, 1975.
Introduction
Lines 40-41: Please remove Fig1A-C from the introduction section as this is a result
Lines 54-55: Please move the Table 1 to the results section as this is a result from your review.
Figure 2: Please add labeling to show the two specimens as it is not easy to discriminate them.
Figure 4: It would be nice to see also the original image from the light microscopy if it is available
Figure 6: Do the colours indicate the gender? If yes, please indicate this in the figure legend
Table 2: In this table the male size is presented as a ratio. I would suggest to the authors to provide the total body length with metric units.

Micro-CT datasets are available through Morphosource. Please provide these details in the main text.

Experimental design

The authors provided new insights in the attachment mechanism of shiinoids and resolved several taxonomic issues regarding the specific species. However, I would suggest to the authors to emphasize this knowledge gap in the introduction section.

Regarding the micro-CT protocol that the authors used, some details about the specimen preparation (e.g. specimen fixation and stored in which solution prior to staining) must be mentioned. Regarding the PTA solution (lines 82-83) did you renew this every day or did you submerge the specimens once?
Details about the softwares used for the micro-CT images should be mentioned (reconstruction, visualisation,segmentation softwares).

Validity of the findings

Data have been provided but need to be cited in the manuscript.
Conclusions are well written and appropriately stated.

·

Basic reporting

This is an excellent study on a fascinating and poorly known group of shiinoid parasitic copepods. The general reporting in scientific style and form is good, the descriptions are clear and easy to follow despite the complexity that comes with taxonomic terminology. I have made some detailed suggestions to possibly improve some clarity and language, although I need to state that I am not a first language English speaker.

The article structure is good and the data presentation and of course the microCT image figures are excellent.

The findings are well presented and the the conclusions are sound.

Experimental design

The study is mainly descriptive and observation.

When it comes to stating the research question and filling knowledge gaps I felt that the introduction could potentially be strengthened through more contextualisation of how this study helps to improve the knowledge within this group and for crustaceans as whole. This is not even a criticism as it merely seemed that the authors, who are so well into their topic, know this and take it for granted without actually writing it down. I would like to encourage the authors to perhaps revisits the manuscript with this in mind to make the paper stronger.

The study is conducted meticulously and the methods are described well.

Validity of the findings

This is a great study that significantly enhances the knowledge on a peculiar and intriguing parasitic group of copepod crustaceans, the shiinoids. Applying microCT to this group is a great and novel approach.

The data attained are well structured and robust. The study is descriptive and the observations are well analysed and support the conclusions.

Additional comments

Once more I'd like to congratulate the authors on a fabulous study. Some additional and more detailed comments that may or may not be helpful to improve the manuscript are listed below.

Abstract

Line 21. Not a full sentence? The antennae form..?
Line 23. in situ
Line 8. The mode of attachment is clearly not novel? It just hasn’t been studied? Please make clear that while you study might have novel elements, the attachment mode evolved with this group.
Line 25 following. The use of semicolons here does not seem to be grammatically working very well? The listing should be rather separated by commas?

Introduction

Line 47. in situ see above. I also suggest to restructure the sentence and start with: To characterize the muscles used…. we…

I acknowledge that very little is known about shiinoids as a group and that the nature of this research is descriptive. However, it feels to me that the introduction could benefit a lot by outlining why this research is important and what implications the insights into shiinoid morphology have for our general understanding of copepod/crustacean morphology or potentially on parasitology as a whole. Why does this research have to be conducted and how does it fit into the bigger picture outside of simply describing the grasping mechanism in this very specialised group? The introduction gives very little context even if this was just on how rare/unstudied these organisms are. It also does not cite a single reference to what is known other than the authorities of the species themselves.

Methods

Line 75 perhaps refer to nasal lamella samples rather than specimens here?
Line 80 perhaps add that males are attached to the female?
Line 83 perhaps write reverse osmosis out

Results
Line 92. Sentences reads strange with the inserted comma. Also use past tense for results. Perhaps … Adult females attached to the nasal lamellae of Scomberomorus spp. (fig..) using a clasp-like mechanism…

Line 184-188. I find this is misplaced in the results section and should go into the introduction and perhaps picked up again in the discussion.

Taxonomic results
Line 215: Replace “round the head” by “near” the head?

220 and below: Is there any reason why you call it a “Differential Diagnosis” and not just “Diagnosis”? Any diagnosis should be differential. I am aware that this is a new one that now differentiates an incorporation of more species, but that is very clear and the nature of any new work.

Line 268: described the female of?


Conclusions:

I always feel that for an article like this a conclusion is merely unnecessary repetition making a manuscript longer than needed.

References:

Line 526: non capital
534 and 558: same

Table 1. Shiinoa inauris second line is “Surinam, Brazi” correct?

---

## Round 0.2 · accepted · Accept

The authors have attended to the reviewers comments in a comprehensive and satisfactory manner. The manuscript is accepted, although there may still be some editorial changes required.